# Association of Cognitive Polygenic Index and Cognitive Performance with Age in Cognitively Healthy Adults

**DOI:** 10.3390/genes14091814

**Published:** 2023-09-18

**Authors:** Angeliki Tsapanou, Margaret Gacheru, Seonjoo Lee, Niki Mourtzi, Yunglin Gazes, Christian Habeck, Daniel W. Belsky, Yaakov Stern

**Affiliations:** 1Department of Neurology, Columbia University Irving Medical Center, New York, NY 10032, USA; at2859@cumc.columbia.edu (A.T.); ch629@cumc.columbia.edu (C.H.); 2Department of Biostatistics, Columbia University Mailman School of Public Health, New York, NY 10032, USA; mg3861@cumc.columbia.edu; 3Department of Psychiatry, Columbia University Irving Medical Center, New York, NY 10032, USA; 4Department of Neurology, National and Kapodistrian University of Athens, 10679 Athens, Greece; 5Department of Epidemiology and Butler Columbia Aging Center, Columbia University Mailman School of Public Health, New York, NY 10032, USA

**Keywords:** polygenic index, cognition, normal aging

## Abstract

Genome-wide association studies have discovered common genetic variants associated with cognitive performance. Polygenic scores that summarize these discoveries explain up to 10% of the variance in cognitive test performance in samples of adults. However, the role these genetics play in cognitive aging is not well understood. We analyzed data from 168 cognitively healthy participants aged 23–77 years old, with data on genetics, neuropsychological assessment, and brain-imaging measurements from two large ongoing studies, the Reference Abilities Neural Networks, and the Cognitive Reserve study. We tested whether a polygenic index previously related to cognition (Cog PGI) would moderate the relationship between age and measurements of the cognitive domains extracted from a neuropsychological evaluation: fluid reasoning, memory, vocabulary, and speed of processing. We further explored the relationship of Cog PGI and age on cognition using Johnson–Neyman intervals for two-way interactions. Sex, education, and brain measures of cortical thickness, total gray matter volume, and white matter hyperintensity were considered covariates. The analysis controlled for population structure-ancestry. There was a significant interaction effect of Cog PGI on the association between age and the domains of memory (Standardized coefficient = −0.158, *p*-value = 0.022), fluid reasoning (Standardized coefficient = −0.146, *p*-value = 0.020), and vocabulary (Standardized coefficient = −0.191, *p*-value = 0.001). Higher PGI strengthened the negative relationship between age and the domains of memory and fluid reasoning while PGI weakened the positive relationship between age and vocabulary. Based on the Johnson–Neyman intervals, Cog PGI was significantly associated with domains of memory, reasoning, and vocabulary for younger adults. There is a significant moderation effect of genetic predisposition for cognition for the association between age and cognitive performance. Genetics discovered in genome-wide association studies of cognitive performance show a stronger association in young and midlife older adults.

## 1. Introduction

Cognitive performance is significantly influenced by genes, with approximately half of the variance in general cognition attributed to genetic factors [1]. Apart from general cognition, distinct cognitive domains are also influenced by genetics; mostly attention, working memory, and declarative memory [2,3]. Pietropaolo and Crusio reported the two main strategies to study genetics of human cognition; the candidate gene approach, and the whole-genome approach [4]. Most of the existing literature focused on the genetics of cognition and Alzheimer’s disease [5], has used the first approach, studying genes whose function has been shown to influence any neurobiological process involved in the phenotype of interest. GWAS analyses have discovered large numbers of small-effect variants [6]. In order to understand how these variants affect cognitive aging, there is a need to follow-up with respondents with more detailed phenotyping. Thus, to address the power problem, researchers have begun to combine GWAS discoveries into a single index. Creation of Polygenic Indices (PGI) is the best method of applying this second approach [7].

In a previous study [8], we found that a Cognitive PGI (Cog PGI) was associated with a summary score of cognitive performance in adults aged 23 to 77 years old. This result remained significant even after including brain markers (cortical thickness, total gray matter volume, and white matter hyperintensity) as covariates. The trajectory of changes in cognition with aging varies by cognitive domain. For example, memory declines with aging both in normal aging and neurodegenerative diseases [9], while language performance remains stable or improves with age [10]. Therefore, in the current analysis we examined the effect on age on the associations between Cog PGI and performance in discrete cognitive domains (fluid reasoning, memory, vocabulary, and speed of processing), including brain measures as covariates. 

## 2. Methods

### Participants

Participants were drawn from two ongoing studies at Columbia University Irving Medical Center: the Reference Ability Neural Network (RANN) study and the Cognitive Reserve (CR) study [11,12]. In the RANN Study, 12 cognitive neuroimaging tasks are administered to healthy adults in order to identify the neural networks associated with four reference abilities (memory, fluid reasoning, speed of processing, and vocabulary) and investigate how these networks are affected by aging. CR was designed to help uncover neural mechanisms that might underlie the cognitive reserve. Participants were required to be native English speakers, right-handed, and have at least a fourth-grade reading level. Screening was performed to ensure that participants had no neurological or psychiatric conditions, cognitive impairment, or MRI contraindications. Older participants who met diagnostic criteria for mild cognitive impairment (MCI) and dementia were excluded. More detailed information about the studies can be found elsewhere [11,12]. Participants included in the final sample had complete data on the polygenic risk scores, cognitive performance in the four domains, and socio-demographic variables. For this study, we considered 168 individuals aged 23–77 whose ethnicity was European-American only.

## 3. Cognitive Tasks

Twelve measures from a battery of neuropsychological tests were selected to assess functioning in four cognitive domains. Based on a principal axis factor analysis, the composite scores for each of the four cognitive domains were determined by specific sets of tasks. The score for fluid reasoning was determined by the following: Wechsler Adult Intelligence Scale (WAIS-III) Block design task, WAIS-III Letter–Number Sequencing test, and WAIS III Progressive Matrices. Processing speed was estimated from WAIS-III Digit Symbol Subtest, Part A of the Trail Making Test, and the Stroop Color Naming tests. The composite score for vocabulary was based on the vocabulary subtest from WAIS-R, the Wechsler Test of Adult Reading (WTAR), and the American National Adult Reading Test (AMNART). Lastly, Episodic memory was calculated from three sub-scores of the Selective Reminding Task (SRT): Long-Term Storage sub-score (SRT LTS), Continuous Long-Term Retrieval (SRT CLRT), and Last Trial (SRT Last). Z-scores were computed for participants, on each summary score, based on the overall means and standard deviations. A higher score indicates better cognitive performance.

The GWAS used for PGI calculation included participants of European ancestry, and previous studies have shown that PGIs perform worse when applied to other ancestry groups due to differences in Linkage-disequilibrium (LD) and allele frequencies between different populations. Thus, we restricted our analysis to self-reported European-American participants.

## 4. PGI Calculation

Genotyping: A venous blood draw was taken for every participant during their appointment at Columbia University. DNA samples were obtained through whole-blood extraction. Genotyping was performed using Omni 1M chips, based on Illumina procedures. Genotype calling was performed using GenomeStudio v.1.0 (https://emea.support.illumina.com/array/array_software/genomestudio/downloads.html, accessed on 1 June 2022). Quality control was applied to both DNA samples and SNPs. Specifically, samples were removed from further analyses if they had call rates below 95%, sex discrepancies, and relatedness (kinship coefficient more than 0.125). To account for population structure-ancestry, we computed the top 20 Principal Components (PCs) of the whole sample using Plink software 1.9 and we used the 20 PCs as covariates in our analyses [13]. 

GWAS imputation: GWAS data were imputed using the Haplotype Reference Consortium (HRC v1.1) panel through the Michigan Imputation online server (Das, Forer et al., 2016). The HRC is a reference panel of 64,976 human haplotypes at 39,235,157 SNPs constructed using whole-genome sequence data from 20 studies of predominantly European ancestry [14].

Imputed dosages for a total of 6,280,331 SNPs with MAF > 0.05, HWE *p*-value > 10^–6^, and a missing rate < 10% were used for PGI computation. PGI scoring was performed using PRSice-2 software [15] following the clumping and thresholding (C + T) approach, as previously described by the International Schizophrenia Consortium [16].

Polygenic Index: We composed the PGI from summary statistics from a recent GWAS meta-analysis of cognitive performance including n = 269,867 participants, from 14 independent European cohorts [8]. Different measures of intelligence were assessed in each cohort but were all operationalized to index a common latent *g* factor, the general intelligence factor or Spearman’s *g*, representing multiple dimensions of cognitive functioning. The majority of the samples were adults, 18 to 60 years old (n = 204,228), and when the participants were stratified according to age groups (children, young adults, older adults, adults), results did not indicate any specific age-dependent effect, suggesting that the same SNPs are significant across age-groups.

For the purposes of the current analysis, all SNPs were included, regardless of *p*-value in the mass-univariate screen. In order to ensure that only independent markers were included in the computed PGI, we conducted LD clumping using an R^2^ threshold of 0.1 and a 250 kb sliding window. Markers within the Major Histocompatibility Complex (MHC) LD region on chromosome 6 (chr6:27–33 Mb, hg19) were also excluded from the PGI due to the presence of complex patterns of long-range linkage disequilibrium within this region. For each remaining SNP, we computed the weighted count of cognition-associated alleles (0, 1, or 2), with the weights determined by the coefficient estimated in the GWAS that corresponds to the effect allele. We then computed the average weighted count across all SNPs to form the PGI. The PGI computation was performed using the PRSice software [15]. For interpretation reasons, PGI values were normalized across our sample by z-transformation. 

## 5. Brain Measures

Structural MRI scan and image processing: MRI images were acquired on a 3.0 T Philips Achieva Magnet. Each scan used a 240 mm field of view. The parameters for EPI acquisition were TE/TR (ms) 20/2000; Flip Angle 72°; In-plane resolution (voxels) 112 × 112; Slice thickness/gap (mm) 3/0; Slices 41. T1 scans for each participant were reconstructed with the FreeSurfer (v5.1.0), which is a software for human brain imaging analysis (http://surfer.nmr.mgh.harvard.edu, (accessed on 1 June 2022)). The accuracy of FreeSurfer’s subcortical segmentation and cortical parcellation [17,18] has been reported to be comparable to manual labeling. Each participant’s white and gray matter boundaries, as well as gray matter and cerebral-spinal-fluid boundaries, were visually inspected slice by slice, and control points were added manually wherever there was a visible discrepancy. Boundary reconstruction was repeated until we obtained satisfactory results for every participant. The subcortical structure borders were plotted by the TkMedit visualization tools and they were compared against the actual brain regions. In case of any discrepancies, we corrected them manually.

Based on our previous publication [8], we selected three neural phenotypes for analysis based on published associations with cognitive test performance: gray matter volume (GM) [19], cortical thickness (CT) [20,21], and white matter hyperintensities (WMH) [22]. We used total gray-matter volume as reported by FreeSurfer. CT was computed as the average of both hemisphere averages provided by standard FreeSurfer parcellation [17]. FreeSurfer’s subcortical segmentation and cortical parcellation have been shown to have comparable accuracy to manual labeling [18,23]. Reconstructions were initially checked and, manual control points and editing were used wherever needed. WMH was measured as follows. First, WMH was segmented by the Lesion Segmentation Tool algorithm (LST) [24] as implemented in the LST toolbox version 2.0.15 (June 2017) for Statistical Parametric Mapping (SPM) (www.statistical-modelling.de/lst.html, (accessed on 1 June 2022)). Next, in order to extract lobar WMH values, we registered the T1 sequence for each participant in FreeSurfer and then co-registered the FLAIR sequence. We log-transformed global WMH burden (log(WMH + 1)) derived from the FMRIB software 6.0 library [25]. The following brain regions were characterized as the lobes in FreeSurfer: frontal, parietal, temporal, cingulate, and occipital. Finally, the volumes of WMH in each of the five lobes were automatically extracted, and their sum was used in the current study. One participant was excluded because of technical problems while extracting the regional WMH map [26].

## 6. Statistical Analysis

We performed the analyses using the statistical package R, version 4.2.1 (https://www.r-project.org (accessed on 23 June 2022)). Statistical significance was defined as a *p*-value < 0.05 using a two-tailed test. We aimed to test the interaction effect of Cog PGI on the relationship between age and each cognitive domain. This was accomplished using linear regression models with the moderator (Cog PGI), age, and their interactions as independent variables. For the models in which the interaction term was significant, the interplay between Cog PGI and age on cognition was examined using the Johnson–Neyman Technique for two-way interactions [27,28]. The Johnson–Neyman interval represents a range of values of the moderator in which the slope of the predictor is significant vs. non-significant at a specified level. All regression models controlled for the following covariates: education, sex, and the first οf 20 principal components to control for potential population sub-structure. In a subsequent model, brain measures (gray matter volume, cortical thickness, and white matter hyperintensities) were included as additional covariates. All continuous independent variables were mean-centered and standardized. 

## 7. Results

Table 1 presents a summary of baseline demographics and cognitive measures of interest. The participants had an average of a bit over 16 years of education and had an average estimated IQ of 120.28. The sample was majority female.

After adjusting for education, sex, and first PC, Cog PGI moderated the relationship between age and cognitive ability for memory (β = −0.158, CI = [−0.292, −0.023], *p*-value = 0.022), fluid reasoning (β = −0.146, CI = [−0.269, −0.024], *p*-value = 0.020), and vocabulary (β = −0.191, CI = [−0.307, −0.076], *p*-value = 0.001) (Table 2). Figure 1 provides a visual representation of the regression coefficients by examining the relationship between age and cognitive ability at three values of Cog PGI (1 SD below the mean, mean, and one SD above the mean). For memory, the negative relationship between age and memory strengthened with higher Cog PGI, meaning that higher values of Cog PGI showed more decline in cognition for each unit increase in age. For fluid reasoning, the negative relationship between age and reasoning also strengthened with a higher Cog PGI. In other words, the decline in reasoning due to age was greater as Cog PGI increased. On the other hand, regarding vocabulary, as the level of Cog PGI increased, the positive relationship between age and performance weakened. Age was positively associated with vocabulary for lower Cog PGI, but there was almost no association for higher Cog PGI. Additionally, we determined the age ranges in which Cog PGI had a significant association with cognition using the Johnson–Neyman Technique. These are denoted by the shaded areas in Figure 1. Higher Cog PGI was associated with better cognitive performance in memory for young adults. Higher Cog PGI was associated with better cognitive performance in fluid reasoning for young and middle-aged adults. Furthermore, Cog PGI had a positive relationship with vocabulary for young and middle-aged adults. 

Once the brain measures were adjusted in the regression model, the interaction between age and Cog PGI on cognition was still observed (Table 3). Although the effect size slightly decreased, Cog PGI significantly moderated the relationship between age and cognitive ability in fluid reasoning, memory, and vocabulary. Adding the brain measures in the model did not alter the overall patterns found in the relationship between age/Cog PGI and cognition.

## 8. Discussion

We tested whether the polygenic index (PGI) moderated the relationship between age and performance on neuropsychological tests of four specific domains of cognitive functioning (memory, fluid reasoning, speed of processing, and vocabulary) in a sample of adults ranging in age from 23 to 77. We found that the general-cognitive-ability PGI was a moderator between age and three cognitive domains—memory, fluid reasoning, and vocabulary. Higher levels of the PGI displayed a greater negative association between age and performance in memory and fluid reasoning while higher levels of PGI weakened the positive relationship between age and vocabulary. Furthermore, the association of PGI with memory, reasoning, and vocabulary was stronger in younger adults and weaker in older adults, suggesting that processes of cognitive aging disrupt the connection between neuropsychological test performance and genetics, as represented by the PGI for general cognitive ability PGI. 

The stronger association of PGI with cognitive domains in younger and middle-aged adults suggests that genetic influences on cognition may vary with age. It is possible that age-related differences in the association between PGI and cognitive domains might reflect underlying developmental processes. Cognitive abilities typically undergo significant changes during early and middle adulthood, with different abilities peaking at different stages. In particular, fluid reasoning tends to peak in early adulthood [29] while vocabulary skills may continue to develop and improve throughout the lifespan [30]. Thus, the observed stronger association between PGI and cognitive performance in younger and middle-aged adults could reflect the influence of genetic factors during critical periods of cognitive development.

The link between PGI and cognitive performance was weaker in advanced ages, even after adjusting for brain measures, suggesting that the contribution of other factors, such as environmental/lifestyle factors, is more important in maintaining normal cognitive function as we get older. Importantly, our cohort consisted of cognitively healthy individuals and as such, the PGI effect on performance was not confounded by the appearance of age-related disorders. However, our results could be because of a subclinical age-related change, where cognitive symptoms are not distinct yet. There are a number of factors that could explain the weaker association of PGI in older ages. First, other genes may become more influential on cognition as we age. Secondly, as we grow age-related pathology accumulates in the brain that may not surpass the critical threshold for appearance of cognition-related disorder but still affect cognitive performance during normal aging [31]. Moreover, genetic variation influencing cognitive abilities may contribute to the lifelong cultivation of cognitive reserve that may modify the relationship between brain pathology and expected cognitive performance [32]. Thus, future longitudinal studies with available neuroimaging data and cognitive evaluation should examine these hypotheses for further insight into the etiology and causation of the link among cognition, genetics, and brain aging. However, it could also be that the attenuated association in older people reflects a kind of selection bias wherein people whose genes contribute to poor cognitive performance are screened out. 

Genes associated with cognition are highly expressed in the brain and are implicated in nervous-related pathways and synaptic structure [33]. The expression profile of cognitive-related genes is altered during aging and these changes may affect the overall functioning of the brain and contribute to age-related changes [34]. Some genes involved in synaptic plasticity, neurotransmission, and neuronal function may be downregulated, while others associated with neuroprotection and repair mechanisms may be upregulated [34,35]. Likewise, the changes observed in the brain during aging could also interact with the mechanisms of PGI, potentially modifying its impact on cognitive abilities. 

The present study has significant strengths. First, it uses not only a general, short cognitive scale for the cognitive evaluation of the participants, but it includes an extensive neuropsychological battery. Thus, the categorization into cognitive domains is easier, which provides specific information about the cognitive status of each participant. Further, the use of brain measures was another strength of the study, since it provides insight into the association between cognition and genetics in normal aging. However, the small sample size of our study was the main limitation. 

Understanding the influence of genetics on cognition in a domain-specific and time-dependent manner is crucial in gaining insights into the mechanisms influencing cognitive performance. Furthermore, the PGI holds the potential to serve as a valuable tool for identifying individuals who are at a heightened genetic risk of experiencing cognitive decline as they grow older. Early identification of individuals more susceptible to age-related cognitive disorders, may allow the implementation of preventive measures and targeted interventions at an earlier stage. This may include lifestyle modifications, cognitive training, or the use of pharmacological interventions to potentially delay or mitigate cognitive decline. Towards personalized medicine, the PGI could aid in developing personalized treatment plans based on an individual’s genetic risk profile. This approach could allow for tailored interventions and therapies that take into account an individual’s specific genetic vulnerabilities, potentially leading to more effective and precise treatment outcomes [36]. Another important implication is in the field of clinical trials where the PGI could assist in stratifying individuals based on their genetic risk and allow researchers to assess the effectiveness of interventions and potential treatments across different risk groups [37]. 

## Figures and Tables

**Figure 1 genes-14-01814-f001:**
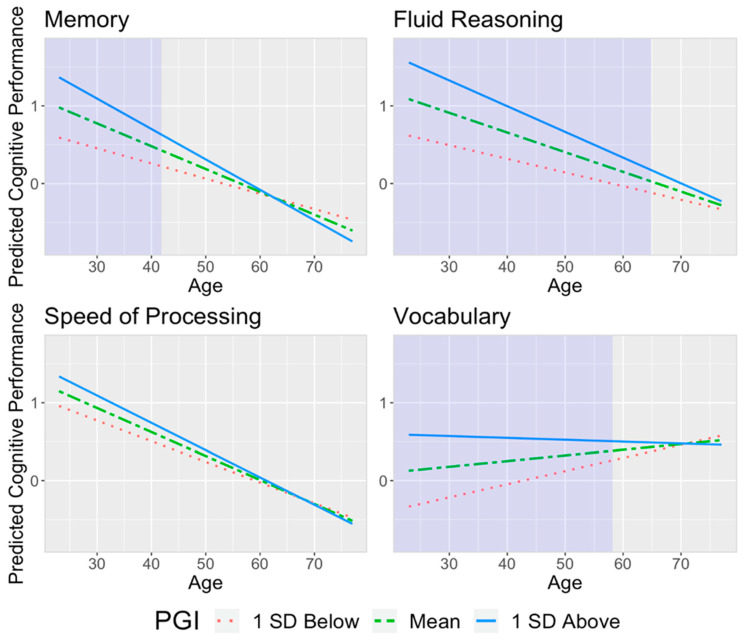
The effect of age on cognitive abilities moderated by Cog PGI, adjusting for sex, education, and first PC. The shaded area represents the Johnson–Neyman interval of ages where Cog PGI had a significant association with cognition.

**Table 1 genes-14-01814-t001:** Baseline characteristics of participants.

Characteristics	N	Mean (SD)	Minimum	Maximum
Age, years	168	56.90 (15.50)	23	77
Sex, N(%)	168			
F		86 (51.19%)	-	-
M		82 (48.81%)	-	-
Education, years	168	16.45 (2.27)	12	22
Cognitive Domains (z−scores)				
Memory	166	−0.02 (0.96)	−2.36	1.60
Fluid Reasoning	168	0.22 (0.79)	−1.39	2.05
Speed of Processing	168	0.10 (0.75)	−1.69	2.10
Vocabulary	165	0.36 (0.69)	−2.03	1.21
Brain Measures				
Cortical Thickness	165	2.53 (0.12)	2.26	2.85
Total Gray Matter Volume	165	625,269.41 (59,112.66)	494,775.94	819,078.57
log(White Matter Hyperintensity + 1)	160	4.74 (2.66)	0	9.50

**Table 2 genes-14-01814-t002:** Regression results for associations between PGI and each cognitive domain cognition with age moderation, adjusting for sex, education, and first PC. Standardized regression coefficients and their 95% confidence intervals are reported.

	(a) Memory			(b) Fluid Reasoning		
Predictors	β	CI	p	β	CI	p
Cog PGI	0.059	[−0.119, 0.237]	0.516	0.253	[0.090, 0.415]	0.003
Age	−0.473	[−0.608, −0.338]	<0.001	−0.478	[−0.601, −0.354]	<0.001
Education	0.246	[0.108, 0.384]	0.001	0.142	[0.017, 0.267]	0.027
Sex [M]	−0.327	[−0.593, −0.060]	0.017	0.108	[−0.138, 0.353]	0.388
PC 1	−0.070	[−0.244, 0.104]	0.427	0.044	[−0.115, 0.204]	0.586
Cog PGI x Age	**−0.158**	**[−0.292, −0.023]**	**0.022**	**−0.146**	**[−0.269, −0.024]**	**0.020**
	**(c) Speed of Processing**			**(d) Vocabulary**		
**Predictors**	**β**	**CI**	**p**	**β**	**CI**	**p**
Cog PGI	0.061	[−0.096, 0.219]	0.444	0.173	[0.017, 0.328]	0.030
Age	−0.635	[−0.756, −0.515]	<0.001	0.144	[0.028, 0.260]	0.015
Education	0.103	[−0.019, 0.225]	0.097	0.322	[0.204, 0.440]	<0.001
Sex [M]	−0.033	[−0.272, 0.205]	0.782	0.063	[−0.169, 0.294]	0.592
PC 1	0.110	[−0.045, 0.266]	0.161	0.029	[−0.122, 0.181]	0.701
Cog PGI x Age	−0.088	[−0.207, 0.031]	0.148	**−0.191**	**[−0.307, −0.076]**	**0.001**

Note: Bold indicates a significant interaction effect between Cog PGI and age on the cognitive domains.

**Table 3 genes-14-01814-t003:** Regression results for associations between PGI and each cognitive domain, with age moderation, adjusting for sex, education, first PC, and brain measures. Standardized regression coefficients and their 95% confidence intervals are reported.

	(a) Memory			(b) Fluid Reasoning		
Predictors	β	CI	p	β	CI	p
Cog PGI	0.082	[−0.105, 0.269]	0.387	0.275	[0.104, 0.446]	0.002
Age	−0.476	[−0.715, −0.236]	<0.001	−0.503	[−0.719, −0.287]	<0.001
Education	0.243	[0.099, 0.387]	0.001	0.119	[−0.011, 0.250]	0.072
Sex [M]	−0.288	[−0.615, 0.040]	0.085	−0.058	[−0.359, 0.242]	0.702
PC 1	−0.096	[−0.276, 0.083]	0.291	0.023	[−0.142, 0.188]	0.784
Cortical Thickness	0.081	[−0.120, 0.281]	0.427	−0.086	[−0.268, 0.096]	0.350
Total Gray Volume	−0.054	[−0.253, 0.145]	0.593	0.176	[−0.007, 0.359]	0.060
log(WMH + 1)	0.057	[−0.142, 0.257]	0.570	0.130	[−0.047, 0.306]	0.149
Cog PGI x Age	**−0.147**	**[−0.288, −0.007]**	**0.040**	**−0.128**	**[−0.256, −0.001]**	**0.049**
	**(c) Speed of Processing**			**(d) Vocabulary**		
**Predictors**	**β**	**CI**	**p**	**β**	**CI**	**p**
Cog PGI	0.068	[−0.102, 0.239]	0.429	0.202	[0.041, 0.363]	0.014
Age	−0.655	[−0.871, −0.440]	<0.001	0.273	[0.073, 0.472]	0.008
Education	0.091	[−0.039, 0.221]	0.167	0.312	[0.192, 0.432]	<0.001
Sex [M]	−0.035	[−0.334, 0.264]	0.818	−0.161	[−0.442, 0.120]	0.259
PC 1	0.099	[−0.065, 0.264]	0.235	−0.015	[−0.168, 0.138]	0.848
Cortical Thickness	0.038	[−0.143, 0.220]	0.677	−0.055	[−0.229, 0.119]	0.536
Total Gray Volume	0.021	[−0.161, 0.204]	0.817	0.238	[0.068, 0.408]	0.006
log(WMH + 1)	0.070	[−0.106, 0.246]	0.436	−0.004	[−0.166, 0.158]	0.959
Cog PGI x Age	−0.096	[−0.223, 0.031]	0.138	**−0.162**	**[−0.279, −0.044]**	**0.007**

Note: Bold indicates a significant interaction effect between Cog PGI and age on the cognitive domains.

## Data Availability

Data are available upon request from the corresponding author.

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
