# Peer review of "Association of Cognitive Polygenic Index and Cognitive Performance with Age in Cognitively Healthy Adults"

_genes, 2023, doi:10.3390/genes14091814_

Round 1

Reviewer 1 Report

While this paper appears interesting at first glance, I recommend that the authors address the following issues before it can be properly reviewed. The reference list appears to be incomplete, as it omits references starting from number 20, despite their citation in the text. Additionally, the authors may want to ensure consistent formatting of references throughout the text.

Author Response

The references are now updated throughout the manuscript.

Reviewer 2 Report

The manuscript entitled “Cognitive Polygenic Index and Cognitive performance: the effect of age” examines a combined effect of polygenic score, age, education, brain imaging measures on four cognitive domains in cognitively healthy adults. Although, such study is interesting, a sample size is rather small.

In addition, there are some points that have to be clarified and corrected:

1. Please, report abbreviations for the PGI in the Abstract.

2. I suggest to change the term “Cognitive genetics” in the Conclusion of the Abstract. In turn, I suggest to include the proportion of variance explained by the regression model in the Abstract.

3. The references for the Reference Abilities Neural Networks, and the Cognitive Reserve study, which summary statistics was used, have to be included in the Abstract.

4. The study also examines the effect of brain-imaging measures on cognitive domains, and the authors have mentioned that this approach was used in their previous study. In this regard, it remains unclear, what is a rationale to conduct the present study comparing with their previous work. In turn, I suggest to mention a relation between brain imaging domains and cognitive functioning, which help the readers to understand such controlling for brain imaging parameters in regression models. I suggest to make changes in the aim of the study, since nothing was reported regarding brain imaging results in the aim.

5. The authors have to explain a rationale for the examination of selected cognitive traits in the Introduction.

6. I have a remark regarding the sample. Please, give the same age range in the Participants subsection and in the Abstract. In addition, please, report mean age and SD, as well as gender content, since sex-related differences in the cognitive domains exist. In this regard, have you controlled for sex in statistical analyses?

7. It is hard to find the correct reference, since the list of reference is incomplete. The authors, have to check their list of references. For example, on line 118 the authors refer to the study [8], which appears to be incorrect. In this regard, it remains unclear, what was the meta-analysis, which summary statistics was used for PGI calculation.

8. Please, provide a reference to R (line 169).

9. It remains unclear if the scores distribution of all examined cognitive subscales coincided to the Gaussian distribution. Please, clarify. In addition, have the authors checked for the possible correlation between brain imaging measures and examined cognitive domains? Was the problem of multicollinearity ruled out?

10. It remains unclear, what score ranges were used for each cognitive domain? Were they standardized? It is hard to explain a negative mean value for the Memory domain in Table 1. I suggest to report a range for the cognitive scores in Materials and Methods section.

11. Please, rephrase the notes to Table 2, since bold reflects not only a significant interaction effect.

12. The authors revealed a negative association between cognitive domains (except for Vocabulary) and age, which is well-established. How can you explain such opposite relation toward Vocabulary?

14. The Results section requires the addition of regression coefficients, p-values in the text, not only in the Tables.

15. The authors have indicated the existence of PGI x age interaction effect on three cognitive domains. However, I suggest that PGI was not a moderator for the Memory, since a significant p-value for the interaction is attributed to a significant effect of age on Memory, while PGI insignificantly affected it (p=0.387).

16. Since the article includes PGI, it requires a figure demonstrating the effect of PGI by deciles on cognitive score. In turn, it is of interest to compare what proportion of variance in cognitive domains is attributed to PGI, age, brain imaging parameters, and their combined effect. The manuscript would benefit from the inclusion of such figure.

17. The authors use the abbreviation PRS, please, explain it in the first appearance in the text.

18. Please, clarify the necessity of the third paragraph in the Discussion or delete it. The authors do not show the effects of certain genes related to nervous-related pathways and their signs of effect on assessed cognitive domains, in this regard, such paragraph is non-relevant for the present study.

19. I suggest adding more discussion with respect to the values of proportion of variance in cognitive domain explained by predictors and various models. Please, compare them with previously published ones.

20. Please, add limitations to the present study, including the small sample size.

21. Finally, the authors have to fill in the required statements regarding Author Contributions, Funding, Informed Consent, Data Availability at the end of the manuscript.

I suggest to accept after major revision, providing that the authors will address the comments above within the main manuscript as required.

Minor English checks are required.

Author Response

The manuscript entitled “Cognitive Polygenic Index and Cognitive performance: the effect of age” examines a combined effect of polygenic score, age, education, brain imaging measures on four cognitive domains in cognitively healthy adults. Although, such study is interesting, a sample size is rather small.

In addition, there are some points that have to be clarified and corrected:

  1. Please, report abbreviations for the PGI in the Abstract.

This is now corrected in the Abstract.

  1. I suggest to change the term “Cognitive genetics” in the Conclusion of the Abstract. In turn, I suggest to include the proportion of variance explained by the regression model in the Abstract.

This is now corrected to: “There is a significant moderation effect of genetic predisposition for cognition for the association between age and cognitive performance”.

  1. The references for the Reference Abilities Neural Networks, and the Cognitive Reserve study, which summary statistics was used, have to be included in the Abstract.

The references are now updated throughout the manuscript.

  1. The study also examines the effect of brain-imaging measures on cognitive domains, and the authors have mentioned that this approach was used in their previous study. In this regard, it remains unclear, what is a rationale to conduct the present study comparing with their previous work. In turn, I suggest to mention a relation between brain imaging domains and cognitive functioning, which help the readers to understand such controlling for brain imaging parameters in regression models. I suggest to make changes in the aim of the study, since nothing was reported regarding brain imaging results in the aim.

Since brain measures were also used in the previous study, we now made it clear that the innovation of the present study is that we examined not only general cognition but cognitive domains in total, and we examined the effect of age on the association between cognitive PGI and cognitive performance. However, based on your comment, we now edited the aim as well: Introduction section, “Therefore, in the current analysis we examined the association between a Cog PGI and performance in discrete cognitive domains (fluid reasoning, memory, vocabulary, and speed of processing). In addition, we examined the effect of age on these associations, including brain measures as covariates.”

  1. The authors have to explain a rationale for the examination of selected cognitive traits in the Introduction.

The cognitive domains used are the gold standard in clinical neuropsychology. References added in the paper show some of the main previous publications using the same traits.

  1. I have a remark regarding the sample. Please, give the same age range in the Participants subsection and in the Abstract. In addition, please, report mean age and SD, as well as gender content, since sex-related differences in the cognitive domains exist. In this regard, have you controlled for sex in statistical analyses?

Sex is included in the analyses; it is reported in the Statistical analysis section, and at the tables as well. The age range is now corrected.

  1. It is hard to find the correct reference, since the list of reference is incomplete. The authors, have to check their list of references. For example, on line 118 the authors refer to the study [8], which appears to be incorrect. In this regard, it remains unclear, what was the meta-analysis, which summary statistics was used for PGI calculation.

The references are now updated throughout the manuscript.

  1. Please, provide a reference to R (line 169).

Reference is now added.

  1. It remains unclear if the scores distribution of all examined cognitive subscales coincided to the Gaussian distribution. Please, clarify. In addition, have the authors checked for the possible correlation between brain imaging measures and examined cognitive domains? Was the problem of multicollinearity ruled out?

In terms of assumptions that need to be met for linear regression, the errors were approximately normally distributed and there were no issues with multicollinearity. To account for a possible relationship between the brain measures and the cognitive scores, brain imaging measures were included in the model as covariates (see Table 3).

  1. It remains unclear, what score ranges were used for each cognitive domain? Were they standardized? It is hard to explain a negative mean value for the Memory domain in Table 1. I suggest to report a range for the cognitive scores in Materials and Methods section.

The values for each cognitive domain are z-scores (centered and standardized). The ranges of these scores are now reported in Table 1

  1. Please, rephrase the notes to Table 2, since bold reflects not only a significant interaction effect.

This is now corrected to: “Bold indicates significant interaction effect between Cog PGI and age on the cognitive domains.”

  1. The authors revealed a negative association between cognitive domains (except for Vocabulary) and age, which is well-established. How can you explain such opposite relation toward Vocabulary?

The positive association between age and vocabulary is expected. Prior studies have shown that, in contract to memory, speed, and reasoning, vocabulary scores increase with age

  1. The Results section requires the addition of regression coefficients, p-values in the text, not only in the Tables.

Regression coefficients are now included in the Results section.

  1. The authors have indicated the existence of PGI x age interaction effect on three cognitive domains. However, I suggest that PGI was not a moderator for the Memory, since a significant p-value for the interaction is attributed to a significant effect of age on Memory, while PGI insignificantly affected it (p=0.387).

PGI is a moderator due to the significance of the interaction term. Our main focus is to evaluate whether the effect of age on cognition is a function of Cog PGI – does the slope of age on cognition change as Cog PGI changes? If the coefficient of the product term is significant, then we have an interaction effect. Looking at an unstandardized model below, the coefficient for age (1) and coefficient for Cog PGI (2) should not be interpreted on their own and their significance is not important.

Cognition = 0 + 1 Age + 2 Cog PGI + 3 Education + … + 4 Age X Cog PGI

There are times where 4 is significant but 1 or 2 is not significant – but we would still conclude that Cog PGI is a moderator. An insignificant 1 would mean that there is no effect of age on cognition when Cog PGI = 0 (but this doesn’t give any information about the slope at other values of Cog PGI). An insignificant 2 would mean that there is no effect of Cog PGI on cognition when age = 0. However, if 4 is significant, this would mean that the slope of age on cognition is changing depending on the value of Cog PGI. The value and significance 1 and 2 only matter on their own when the interaction term is not included in the model.

Looking at the results in Table 2 (with centered and standardized variables):

Memory = – 0.476 Age + 0.082 Cog PGI + 0.243 Education + … - 0.147 Age X Cog PGI

Since the product term has a significant coefficient, we have an interaction effect. Depending on which variable is considered the moderator and predictor of interest, we can have two interpretations:

If age is the predictor and Cog PGI is the moderator (this was our framework):

  1. a) Significant “-0.476 Age”: There is a negative relationship between age and cognition when Cog PGI = mean value
  2. b) Significant “-0.147” on the product term: For higher Cog PGI, the negative relationship between age and cognition is even greater. For example, 1) at Cog PGI = mean – 1SD, the slope of age on cognition is -0.329, 2) at Cog PGI = mean, the slope of age on cognition is -0.476, and 3) at Cog PGI = mean + 1SD, the slope of age on cognition is -0.623

If Cog PGI is presented as the predictor and age as the moderator:

  1. a) Insignificant “0.082 Cog PGI”: There is a no relationship between Cog PGI and cognition when age = mean value
  2. b) Significant “-0.147” on the product term: As age increases, the association between Cog PGI and cognition changes by -0.147. For example, 1) at age = mean – 1SD, the slope of Cog PGI on cognition is +0.229, 2) at Cog PGI = mean, the slope of age on cognition is +0.082, and 3) at Cog PGI = mean + 1SD, the slope of age on cognition is -0.065
  3. Since the article includes PGI, it requires a figure demonstrating the effect of PGI by deciles on cognitive score. In turn, it is of interest to compare what proportion of variance in cognitive domains is attributed to PGI, age, brain imaging parameters, and their combined effect. The manuscript would benefit from the inclusion of such figure.

This figure (cognition on the y-axis and Cog PGI on the x-axis) would require:

  • Making an assumption that Cog PGI is the main predictor – so we can look at the effect of PGI on cognition
  • Using a regression model without the interaction term so we can interpret the main effects and influence of the predictors without complications (as referenced in the previous comment/response)

However, the main aim of this manuscript is to evaluate whether Cog PGI moderates the relationship between age and cognition, i.e., age is treated as the main predictor and Cog PGI is treated as the moderator. Additionally, we looked at the model without the interaction term in our analysis. However, this was not the focus of our research question and since we found significant interaction effects, so it was excluded from the manuscript. If you or the editor still want to include this in the manuscript, we are more than glad to do so.

  1. The authors use the abbreviation PRS, please, explain it in the first appearance in the text.

Apologies for the confusion. The terms are quite the same; Polygenic Risk Score (PRS), Polygenic Index (PGI). However, since the PGI is the most precise and up-to-date term to use, we stick with this one. It is now consistent throughout the manuscript.

  1. Please, clarify the necessity of the third paragraph in the Discussion or delete it. The authors do not show the effects of certain genes related to nervous-related pathways and their signs of effect on assessed cognitive domains, in this regard, such paragraph is non-relevant for the present study.

This paragraph is now deleted.

  1. I suggest adding more discussion with respect to the values of proportion of variance in cognitive domain explained by predictors and various models. Please, compare them with previously published ones.

The discussion section is now reedited.

  1. Please, add limitations to the present study, including the small sample size.

This is now added, Discussion section: “The present study has significant strengths. First, it uses not only a general, short cognitive scale for the cognitive evaluation of the participants, but it includes an extensive neuropsychological battery. Thus, the categorization into cognitive domains is easier, which provides specific information about the cognitive status of each participant. Further, the use of brain measures was another strength of the study, since it provides insight on the association between cognition and genetics in normal aging. However, the main limitation of the study that should also be noted is the small sample size.”

  1. Finally, the authors have to fill in the required statements regarding Author Contributions, Funding, Informed Consent, Data Availability at the end of the manuscript.

These are now added in the manuscript.

I suggest to accept after major revision, providing that the authors will address the comments above within the main manuscript as required.

Minor English checks are required.

Reviewer 3 Report

General comments:

In the manuscript, the authors tested the association between a cognitive polygenic index and performance in four discrete cognitive domains (fluid reasoning, memory, vocabulary, and speed of processing) in different age groups, on a sample of 168 cognitively healthy participants aged 23 to 77 years. The study is properly conducted, English is good, but the manuscript needs some improvements.

The introductory part and the materials and methods are completely clearly written, however, the matter gets complicated when interpreting the results. In lines 188-189, the authors wrote that Cog PGI moderated the relationship between age and three tested cognitive abilities (memory, reasoning and vocabulary). Please explain in more details your findings from Table 2, because the entire later text of that section refers to the interpretation of Figure 1. I find it strange that the authors decided to use the verb "moderate" to describe the effect on the three cognitive domains, given that the beta values ​​for age in the Memory and Reasoning domains are already individually negative and statistically significantly related to these two domains, while in the Vocabulary domain is beta value for age positive and only by multiplying the variables Cog PGI & age does it get a negative value?!

I would say that the interaction of Cog PGI and age, adjusted for other variables, moderates the decline in Memory and Reasoning with aging compared to the decline in Memory and Reasoning associated with age alone. On the other hand, if we look at Cog PGI and age separately, vocabulary improves with age, while their interaction shows that vocabulary deteriorates with age.

Also, I hope that in Table 2 you did not include the individual Cog PGI and age variables together with their product variable in the same regression model.

When describing the results in Figure 1, you wrote that for memory (lines 195-196), „the negative relationship between age and memory strengthened with a high level of Cog PGI“. What exactly do you mean when you say that the relationship „strengthened“- does this statement refer to a drop in the absolute value of predicted cognitive performance from age 23 until cca age 42, which is greater for +1SD than for mean or -1SD? If so, please describe your results that way – make the explanation as simple as possible. Plus, it sounds very strange to use term high „level“ of Cog PGI – I would just say higher Cog PGI.

Please change the Discussion in accordance with these suggestions (avoid using the terms strengthened or weakened).

Specific comments:

The title could be better, it is not clear what exactly the paper is about. I suggest something like „Association of cognitive polygenic index and cognitive performance with age in cognitively healthy people“

line 4 – as defined in the Journal`s Instructions for Authors „Authors' full first and last names must be provided“. The numbers that define affiliations should be in superscript.

Again, as defined in the Journal`s Instructions for Authors, „The abstract should be a single paragraph and should follow the style of structured abstracts, but without headings“. Thus, please delete „Background:“, „Methods:“, „Results:“ and „Conclusion:“ from the abstract.

In line 16, the authors wrote that the study participants were 20-80 years old, while the Materials and Methods section states that the age range was 23-77. Please correct it.

https://www.mdpi.com/journal/genes/instructions

lines 33-34 Please rephrase this sentence to „ … cognitive performance show stronger association in young and midlife adults than in older adults.“

line 48 – please change the word „samples“ to „subjects“ or „respondents“

There are serious problems in the Materials and Methods section.

1) In the first sentence of the Methods, the authors state that their sample is from the RANN and CR studies, and they refer to references 1 and 2 (lines 62-64). I checked the descriptions of the respondents in those two references; in the first one, research was conducted on identical and fraternal twins from Ohio, Minnesota, and Colorado, while the subjects in the second study were from Philadelphia, ages 8-21.

2) In lines 76 and 95 you mentioned „white ethnicity“ and „white participants“ – please replace the term „white“ with „European American“.

3) In the first section of the subtitle Cognitive Task, the authors mentioned numerous scales and tests, but did not cite their authors such as Wechsler (1999), Grober and Sliwinski (1991), Buschke and Fuld (2011).

4) This paper has 19 works cited in the list of references. And yet, in the 95th line the authors refer to references 21-23, in the 106th line to reference 24, in the 111th line to reference 25, in the 114th and 134th lines to reference 26, and in the 115th line to reference 27.

5) The studies cited in lines 109 and 150 are not listed in the Reference list.

Tables 1, 2 and 3 - it seems as if all the cells were created by merging two rows - I think that displaying the data in only one row per cell would allow for example Table 3 to be on the same page.

In Table 1, please add that you are showing z-values of cognitive domains. Measuring units for cortical thickness and total gray matter volume are missing.

The lines in Figure 1 are shown in three different colors, however when printed in black and white, it is not clear which line is which. Can you please display the line charts so that each line has its own pattern (solid, dot, dash, etc)? And please explain what the markers on the lines represent.

There is no need for names and descriptions of tables and figures written in a smaller font than in the rest of the manuscript. The description of Table 1 is given in bold, while the descriptions of the other tables are not.

line 231 – a letter d is missing in „were associated“

line 239 – this is the first time you use the abbreviation PRS - firstly, you didn't explain its full name, and secondly, why after consistently using the term PGI did you suddenly start using PRS in this section?

line 240 – here you wrote  „supported that …. may vary …“ –  to support as a verb has too strong a meaning to be followed by „may“, please change it to „suggests“

line 246 – space is missing after period

line 247 – please change the term „middle adults“

line 250 – please change „nervous-related“ to „neural“

line 251 – please change „cognitive-related“ to „cognition-related“

line 260 – there is an extra space after period

line 260 – please change „cognitive healthy individuals“ to „cognitively healthy individuals“

Please write if there are any limitations of your work.

The „Back matter“ part of the manuscript was not prepared according to the Journal`s instructions – there are no Author Contribution section, nor Institutional Review Board Statement, Informed Consent Statement, Data Availability Statement and Conflicts of Interest statement.

https://www.mdpi.com/journal/genes/instructions

Author Response

General comments:

In the manuscript, the authors tested the association between a cognitive polygenic index and performance in four discrete cognitive domains (fluid reasoning, memory, vocabulary, and speed of processing) in different age groups, on a sample of 168 cognitively healthy participants aged 23 to 77 years. The study is properly conducted, English is good, but the manuscript needs some improvements.

The introductory part and the materials and methods are completely clearly written; however, the matter gets complicated when interpreting the results. In lines 188-189, the authors wrote that Cog PGI moderated the relationship between age and three tested cognitive abilities (memory, reasoning and vocabulary). Please explain in more details your findings from Table 2, because the entire later text of that section refers to the interpretation of Figure 1.

Figure 1 and Table 2 go hand in hand. Figure 1 provides a visual representation of the findings in Table 2. The description of the figure is meant to provide a more intuitive description of the numbers in the table. 

I find it strange that the authors decided to use the verb "moderate" to describe the effect on the three cognitive domains, given that the beta values ​​for age in the Memory and Reasoning domains are already individually negative and statistically significantly related to these two domains, while in the Vocabulary domain is beta value for age positive and only by multiplying the variables Cog PGI & age does it get a negative value?!

The individual “main” effect should not be interpreted on its own. Please see response below.

I would say that the interaction of Cog PGI and age, adjusted for other variables, moderates the decline in Memory and Reasoning with aging compared to the decline in Memory and Reasoning associated with age alone. On the other hand, if we look at Cog PGI and age separately, vocabulary improves with age, while their interaction shows that vocabulary deteriorates with age.

Please see below for a detailed explanation on the interpretation of the coefficients.

Also, I hope that in Table 2 you did not include the individual Cog PGI and age variables together with their product variable in the same regression model.

The focus of this paper was to evaluate the interaction effect of Cog PGI and age on cognition. In other words, does the slope of age on cognition differ by value of Cog PGI? Therefore, Cog PGI, age, and their product are included in the model (unstandardized version):

Cognition = 0 + 1 Age + 2 Cog PGI + 3 Education + … + 4 Age X Cog PGI

In terms of interpretation, we are mostly interested in the coefficient corresponding to the product term (4). With the presence of the product term, 1 and 2 on their own are fairly meaningless. For example, is interpreted as follows: when Cog PGI = 0, then a 1 year increase in age is associated with a 1 decrease/increase in cognition (holding other variables constant). is interpreted as follows: when age = 0, then 1 unit increase in Cog PGI is associated with a 2 decrease/increase in cognition(holding other variables constant). However, we are more interested in evaluating whether the effect of age on cognition is a function of Cog PGI (4). If 4 is significant, this means the slope of age on cognition is changing depending on the value of Cog PGI. Thus, Cog PGI moderates the relationship between age and cognition.

The interpretation is different if we were looking at a model without the interaction term.

Cognition = 0 + 1 Age + 2 Cog PGI + 3 Education …

Without the interaction term, 1 and 2 become meaningful. 1 is the slope of age on cognition, holding other variables constant. But this model does not inform us on whether the slope of age on cognition at differs based on values of Cog PGI. We looked at this model in our analysis. However, this was not the focus of our research question, so it was excluded from the manuscript.

Looking at the regression results in Table 2:

Memory = – 0.473 Age + 0.059 Cog PGI + 0.246 Education + … - 0.158 Age X Cog PGI

The focus is not on “-0.473” for age and “0.059” for Cog PGI – these values don’t tell us much/don’t answer our question. These are just point estimates when Cog PGI/age is set to a specific value. Since our variables were centered and standardized, here is the interpretation of “-0.473” for age: at Cog PGI = mean value, there is a negative relationship between age and cognition (this doesn’t tell us about the slope of age on cognition at the other values of Cog PGI). What we care about is the “-0.158” on the product term: For every unit increase in Cog PGI, the decline of cognition due to age is even greater. For example, 1) at Cog PGI = mean – 1SD, the slope of age on cognition is -0.315, 2) at Cog PGI = mean, the slope of age on cognition is -0.473, and 3) at Cog PGI = mean + 1SD, the slope of age on cognition is -0.631. We observe a similar situation for fluid reasoning.

For vocabulary, we encounter a situation where the beta estimates result in a change in direction of the relationship between age and cognition.

Vocabulary = 0.173 Age + 0.144 Cog PGI + 0.322 Education + … - 0.191 Age X Cog PGI

The “0.173” on age means: at Cog PGI = mean value, there is a positive relationship between age and cognition (this doesn’t tell us anything about the slope at other values of Cog PGI). However, the “-0.191” implies that the increase of cognition due to age is lessened as Cog PGI increases.  For example, 1) at Cog PGI = mean – 1SD, the slope of age on vocabulary is +0.364, 2) at Cog PGI = mean, the slope of age on vocabulary is +0.173, and 3) at Cog PGI = mean + 1SD, the slope of age on cognition is -0.018 (essentially no effect of age for higher Cog PGI)

When describing the results in Figure 1, you wrote that for memory (lines 195-196), „the negative relationship between age and memory strengthened with a high level of Cog PGI “. What exactly do you mean when you say that the relationship „strengthened “- does this statement refer to a drop in the absolute value of predicted cognitive performance from age 23 until cca age 42, which is greater for +1SD than for mean or -1SD? If so, please describe your results that way – make the explanation as simple as possible. Plus, it sounds very strange to use term high „level “of Cog PGI – I would just say higher Cog PGI.

Please change the Discussion in accordance with these suggestions (avoid using the terms strengthened or weakened).

Added more detail to the manuscript. The phrase “strengthened” refers to the fact that the absolute value of the slope of age on cognition is even greater as Cog PGI increases (more negative). “Weakened” means that the slope of age on cognition becomes less steep.

Specific comments:

The title could be better, it is not clear what exactly the paper is about. I suggest something like „Association of cognitive polygenic index and cognitive performance with age in cognitively healthy people “

This is now changed to the suggested one.

line 4 – as defined in the Journal`s Instructions for Authors „Authors' full first and last names must be provided “. The numbers that define affiliations should be in superscript.

This is now added.

Again, as defined in the Journal`s Instructions for Authors, „The abstract should be a single paragraph and should follow the style of structured abstracts, but without headings “. Thus, please delete „Background: “, „Methods: “, „Results: “and „Conclusion: “from the abstract.

Deleted.

In line 16, the authors wrote that the study participants were 20-80 years old, while the Materials and Methods section states that the age range was 23-77. Please correct it.

Corrected.

lines 33-34 Please rephrase this sentence to „ … cognitive performance show stronger association in young and midlife adults than in older adults. “

Changed

line 48 – please change the word „samples “to „subjects “or „respondents “

Changed

There are serious problems in the Materials and Methods section.

1) In the first sentence of the Methods, the authors state that their sample is from the RANN and CR studies, and they refer to references 1 and 2 (lines 62-64). I checked the descriptions of the respondents in those two references; in the first one, research was conducted on identical and fraternal twins from Ohio, Minnesota, and Colorado, while the subjects in the second study were from Philadelphia, ages 8-21.

The references are now updated throughout the manuscript.

2) In lines 76 and 95 you mentioned „white ethnicity “and „white participants “– please replace the term „white “with „European American “.

Although in genetics the term „whites “is used, we now corrected the term as suggested.

3) In the first section of the subtitle Cognitive Task, the authors mentioned numerous scales and tests, but did not cite their authors such as Wechsler (1999), Grober and Sliwinski (1991), Buschke and Fuld (2011).

4) This paper has 19 works cited in the list of references. And yet, in the 95th line the authors refer to references 21-23, in the 106th line to reference 24, in the 111th line to reference 25, in the 114th and 134th lines to reference 26, and in the 115th line to reference 27.

5) The studies cited in lines 109 and 150 are not listed in the Reference list.

Tables 1, 2 and 3 - it seems as if all the cells were created by merging two rows - I think that displaying the data in only one row per cell would allow for example Table 3 to be on the same page.

We reworked on that.

In Table 1, please add that you are showing z-values of cognitive domains. Measuring units for cortical thickness and total gray matter volume are missing.

Added and explained on the Methods section

The lines in Figure 1 are shown in three different colors, however when printed in black and white, it is not clear which line is which. Can you please display the line charts so that each line has its own pattern (solid, dot, dash, etc)? And please explain what the markers on the lines represent.

Made changes to figure 1

There is no need for names and descriptions of tables and figures written in a smaller font than in the rest of the manuscript. The description of Table 1 is given in bold, while the descriptions of the other tables are not.

Corrected.

 line 231 – a letter d is missing in „were associated “

Corrected.

line 239 – this is the first time you use the abbreviation PRS - firstly, you didn't explain its full name, and secondly, why after consistently using the term PGI did you suddenly start using PRS in this section?

Apologies for the confusion. The terms are quite the same; Polygenic Risk Score (PRS), Polygenic Index (PGI). However, since the PGI is the most precise and up-to-date term to use, we stick with this one. It is now consistent throughout the manuscript.

line 240 – here you wrote „supported that …. may vary … “– to support as a verb has too strong a meaning to be followed by „may “, please change it to „suggests “

Changed.

line 246 – space is missing after period

Corrected.

line 247 – please change the term „middle adults “

Changed to “middle-aged „.

line 250 – please change „nervous-related “to „neural “

Based on a comment of the other reviewer, this paragraph is now deleted.

line 251 – please change „cognitive-related “to „cognition-related “

Changed.

line 260 – there is an extra space after period

Corrected.

line 260 – please change „cognitive healthy individuals “to „cognitively healthy individuals “

Changed.

Please write if there are any limitations of your work.

This is now added, Discussion section: “The present study has significant strengths. First, it uses not only a general, short cognitive scale for the cognitive evaluation of the participants, but it includes an extensive neuropsychological battery. Thus, the categorization into cognitive domains is easier, which provides specific information about the cognitive status of each participant. Further, the use of brain measures was another strength of the study, since it provides insight on the association between cognition and genetics in normal aging. However, the main limitation of the study that should also be noted is the small sample size.”

The „Back matter “part of the manuscript was not prepared according to the Journal`s instructions – there are no Author Contribution section, nor Institutional Review Board Statement, Informed Consent Statement, Data Availability Statement and Conflicts of Interest statement.

This information is now added in the manuscript.

Round 2

Reviewer 1 Report

I thank the authors for updating the references.

In this paper, the authors performed an analysis of the associations of a cognitive polygenic index (Cog PGI) with components of overall cognitive performance. This work extends the authors’ previous publication focused on the association of the Cog PGI with a summary score characterizing overall cognitive performance. In general, this is an interesting and well-organized paper.

I have, however, the following concerns:

First, although an approach to construct genome-wide PGIs using all SNPs has been previously advocated [1], it is also clear that this approach has substantial statistical and translational limitations. The statistical limitation is that the majority of the associations—even those in large-scale GWAS—are within the range of stochastic variation. The translational limitation is that it is hard to implicate insights from the genome-wide PGI with translational efforts to health care. Researchers attempt to ameliorate this issue by considering different cutoffs for significance, including genome-wide significance [2, 3]. The authors may strengthen their analysis by evaluating the associations of Cog PGI constructed using SNPs selected at multiple cutoffs.

Second, to calculate Cog PGI, the authors “computed the weighted count of cognition-associated alleles (0, 1, or 2)…” This definition does not explicitly indicate whether a specific allele was designated as the effect allele, such as considering the minor allele as the effect allele. If the authors established the direction of effects instead of designating a specific effect allele, it reinforces my first concern.

Third, although the authors stated that their goal was to evaluate interactions of PGI with age, such analyses usually include the estimates of the main effects in the models without the interaction terms. They are, however, not provided in this paper.

[1] Khera AV, Chaffin M, Aragam KG, Haas ME, Roselli C, Choi SH, et al. Genome-wide polygenic scores for common diseases identify individuals with risk equivalent to monogenic mutations. Nat Genet. 2018;50:1219-24.

[2] Escott-Price V, Shoai M, Pither R, Williams J, Hardy J. Polygenic score prediction captures nearly all common genetic risk for Alzheimer's disease. Neurobiol Aging. 2017;49:214 e7- e11.

[3] Bellenguez C, Kucukali F, Jansen IE, Kleineidam L, Moreno-Grau S, Amin N, et al. New insights into the genetic etiology of Alzheimer's disease and related dementias. Nat Genet. 2022;54:412-36.

Author Response

First, although an approach to construct genome-wide PGIs using all SNPs has been previously advocated [1], it is also clear that this approach has substantial statistical and translational limitations. The statistical limitation is that the majority of the associations—even those in large-scale GWAS—are within the range of stochastic variation. The translational limitation is that it is hard to implicate insights from the genome-wide PGI with translational efforts to health care. Researchers attempt to ameliorate this issue by considering different cutoffs for significance, including genome-wide significance [2, 3]. The authors may strengthen their analysis by evaluating the associations of Cog PGI constructed using SNPs selected at multiple cutoffs.

Thank you for the comment. We chose the specific cut-off to be consistent with our reference paper and our previous publication (Tsapanou et al., 2023). However, if you want to use different cut-off scores, we would be glad to do so in a supplementary analysis.

Second, to calculate Cog PGI, the authors “computed the weighted count of cognition-associated alleles (0, 1, or 2)…” This definition does not explicitly indicate whether a specific allele was designated as the effect allele, such as considering the minor allele as the effect allele. If the authors established the direction of effects instead of designating a specific effect allele, it reinforces my first concern.

It is now mentioned in the Methods section: “For each remaining SNP, we computed the weighted count of cognition-associated alleles (0, 1, or 2), with the weights determined by the coefficient estimated in the GWAS that corresponds to the effect allele.”

Third, although the authors stated that their goal was to evaluate interactions of PGI with age, such analyses usually include the estimates of the main effects in the models without the interaction terms. They are, however, not provided in this paper.

 We had initially added the main effects as well, however, it was a bit confusing for some reviewers regarding the interpretation of the results. For that reason, we decided to leave that part out of the manuscript.

Reviewer 2 Report

The manuscript has been thoroughly updated, all the issues have been clarified. It can be accepted in the present form.

Author Response

Thank you for the revision

Reviewer 3 Report

The authors have refined the manuscript according to the suggestions, so now I propose it for publication.

Author Response

Thank you for the revision